# Carbonyl Compounds Containing Formaldehyde Produced from the Heated Mouthpiece of Tobacco Sticks for Heated Tobacco Products

**DOI:** 10.3390/molecules25235612

**Published:** 2020-11-28

**Authors:** Yong-Hyun Kim, Young-Ji An, Jae-Won Shin

**Affiliations:** 1Jeonbuk Department of Inhalation Research, Korea Institute of Toxicology, Jeongeup 56212, Korea; anyz0522@naver.com; 2Human and Environmental Toxicology, University of Science and Technology, Daejeon 34113, Korea; 3Department of Toxicology Evaluation, Konyang University, Daejeon 35365, Korea; sjw9146@naver.com

**Keywords:** heated tobacco product (HTP), tobacco stick, formaldehyde

## Abstract

Diverse harmful compounds can be emitted during the heating of tobacco sticks for heated tobacco products (HTPs). In this study, the generation of harmful compounds from the filter, instead of tobacco in tobacco sticks, was confirmed. The heat of a heated tobacco product device can be transferred to the tobacco stick filter, resulting in the generation of harmful compounds from the heated filter. Since the heating materials (tobacco consumable) of the tobacco sticks evaluated in this study were different depending on the brand, the harmful compounds emitted from the heated tobacco stick filters were examined by focusing on the carbonyl compounds, using three different tobacco stick parts. Acetaldehyde and propionaldehyde exhibited the highest concentrations in HTP aerosols produced by heating the tobacco consumable (conventional case) (63.5 ± 18.4 µg/stick and 1.71 ± 0.123 µg/stick, respectively). The aerosols produced by heating tobacco stick filters had higher formaldehyde and acrolein concentrations (0.945 ± 0.214 µg/stick and 0.519 ± 0.379 µg/stick) than the aerosols generated from heated tobacco consumable (0.641 ± 0.092 µg/stick and 0.220 ± 0.102 µg/stick). As such, formaldehyde and acrolein were produced by heating small parts of the mouthpiece of a tobacco stick, regardless of the heated tobacco product brand. In addition, acetone was only detected in the aerosols generated from heated filters (0.580 ± 0.305 µg/stick). Thus, safety evaluations of heated tobacco products should include considerations of the harmful compounds generated by heating tobacco stick mouthpieces for heated tobacco products in addition to those found in heated tobacco product aerosols.

## 1. Introduction

The demand for new types of cigarettes, such as electronic cigarettes and heated tobacco products (HTPs), has been growing, and the diversity of available types continues to increase [1,2,3,4]. Many conventional cigarette smokers prefer HTPs because of the high similarity between HTPs and conventional cigarettes (e.g., smoke and cigarette shape) [5]. In the case of HTPs, the inhalable harmful compounds present in the HTP aerosol are determined by the HTP device conditions (e.g., heating temperature, heating duration time, and HTP stick position in the HTP device) as well as smoking habits [6,7]. Thus, to confirm the health effects of inhaling HTP aerosol, the harmful compounds in aerosol generated by HTP devices need to be evaluated under different conditions.

HTP sticks consist of a tobacco consumable and a mouthpiece [8]. The tobacco consumable is heated directly by the heating panel of the HTP device. During heating, the heat can be transferred to other parts of the stick, such as the paper tube, the hollow acetate tube, and the polymer-film filter, and the tobacco. The heat transfer levels can differ depending on the position of the various components in an HTP device and harmful compounds can be generated by heating the plastic filter or other parts of HTP sticks [9].

In this study, HTP aerosol was generated using three major brands of HTP sticks and devices, and the carbonyl compounds in the aerosol were analyzed. As the heating materials in the HTP sticks were different, the emission of harmful compounds from sources other than tobacco was confirmed.

## 2. Results and Discussion

### 2.1. Calibration and Quality Assurance (QA) Results

Calibration data were obtained by high-performance liquid chromatography (HPLC)-UV analysis of the working standards containing six 2,4-dinitrophenylhydrazine (DNPH)-derivatized carbonyls. In particular, the response factor (RF, µL·ng^−1^; RF = slope of the mass (x-axis) vs. peak area count (y-axis) curve), coefficient of determination (R^2^), relative standard deviation (RSD, %), and limit of detection (LOD, pg/µL and ppbv) were determined for each compound.

The RF values of the six carbonyl compounds were in the range of 243,030 (crotonaldehyde (CA)) to 521,019 (formaldehyde (FA)) µL·ng^−1^. All the target carbonyls showed good linearity (R^2^ > 0.99) and reproducibilities with RSDs below 10%, with a mean R^2^ value of 0.9993 ± 0.0015 (n = 6) and a mean RSD value of 2.51% ± 3.28% (n = 6). All carbonyls exhibited low LOD values, with a mean of 0.17 ± 0.04 pg·µL^−1^. Assuming an aerosol sampling volume of 0.33 L, the mean LOD value for the carbonyls was computed as 1.17 ± 0.09 ppbv. As such, the present analytical system is optimal for the quantitation of trace-level carbonyls in HTP aerosol samples (Table 1).

### 2.2. Comparison of Carbonyl Compounds in Different HTP Samples and Brands

All the target carbonyls except crotonaldehyde (CA) were detected in all the samples (Figure 1). In sample A, the formaldehyde (FA) had relatively low concentration (0.138 ± 0.016 µg/stick). Furthermore, sample B had higher FA concentrations (0.945 ± 0.214 µg/stick) than sample C (0.641 ± 0.092 µg/stick), regardless of the HTP brand. The highest concentration of FA was produced by heating the mouthpiece filter parts against the tobacco consumable parts. These results show that FA can be produced by heating not only tobacco but also tobacco stick filters. Similar to FA, acrolein (ACR) was also emitted in higher concentrations in sample B (0.519 ± 0.379 µg/stick) than in samples A and C (0.121 ± 0.109 µg/stick and 0.220 ± 0.102 µg/stick, respectively).

The highest concentrations of acetaldehyde (AA) (63.5 ± 18.4 µg/stick) were observed in sample C. In the case of samples A and B, relatively low concentrations of AA were detected (0.616 ± 0.732 µg/stick and 1.21 ± 0.650 µg/stick, respectively). Thus, AA was mostly generated by heating tobacco. Furthermore, the AA concentration depended on the HTP brand, with a relatively low concentration of AA observed for HTP-2 (Figure 2). The trend observed for the generation of propionaldehyde (PA) in HTP aerosol was similar to that for AA. The mean concentrations of PA in samples A and B were low (0.2 µg/stick), whereas those in sample C were relatively high (1.71 ± 0.123 µg/stick).

Although acetone (AT) was not detected in HTP aerosol (sample C), it was detected in sample B for all the HTPs (HTP-1, -2, and -3) at a concentration of 0.580 ± 0.305 µg/stick. The statistical significance of the carbonyl generation patterns depending on the tobacco stick heating materials and the HTP brands was evaluated using analysis of variance (ANOVA) tests (Table 2). The carbonyl generation patterns of three carbonyls (FA, AA, and PA) with different heating materials (samples A, B, and C) were statistically significant, with p-values less than 0.01 (*p*-value = 9.05 × 10^−4^ (FA), 5.05 × 10^−4^ (AA), and 8.54 × 10^−6^ (PA)). The other carbonyls (ACR and AT) exhibited high *p*-values of 0.179 and 0.131, respectively. In contrast, when the concentrations of the carbonyl compounds in the HTP aerosol samples were evaluated in relation to the HTP brands, the concentration differences were statistically negligible, with *p*-values greater than 0.3 (*p*-values = 0.949 (FA), 0.914 (AA), 0.358 (ACR), 0.540 (AT), and 0.980 (PA)). As such, the statistical results confirmed that the heating materials in the HTP sticks were an important factor in determining the concentrations of three carbonyl compounds (FA, AA, and PA) in HTP aerosol.

The actual heating temperature used for aerosol production from an HTP stick is affected by several variables, including the heat setting value of the HTP device, the heating time, and the heating materials (tobacco, filter, tube, etc.). Uchiyama et al. [10] verified that the HTP aerosol quantity depends on the heating time of the HTP device. In general, although tobacco sticks for HTP heat only the consumable tobacco part, the heat can be transferred to the filter part. In previous studies, volatile organic compounds were shown to be produced by heating the filter [9], and this study shows that FA was generated by heating some parts of the HTP stick filter. In addition, the FA generation patterns are significantly affected by the different heating materials in HTP sticks. Thus, extreme concentrations of FA can be generated depending on the user’s HTP smoking approach and habits (tobacco stick equipped with device, heating time, continuous use, etc.).

In previous studies, carbonyl compounds were detected from HTP aerosols at higher concentrations than occurred in sample C of this study [11,12,13,14]. They generated HTP aerosol using the “Health Canada Intense (ISO intense)” method [15] as in this study, but the puff number was higher than in this study. However, cutting the tobacco plug (as in this study) could have led to a change in the energy absorption pattern of the non-tobacco material during the test compared to an unaltered tobacco stick. The concentrations of carbonyl compounds in HTP aerosols are increased dramatically by continuously increasing the puff number [16]. In addition, the correction of the system blank is important to determine the concentrations of target carbonyl compounds. Forster et al. [12] measured the air/method blank and FA blank concentration as 1.17 ± 0.20 µg/stick. If the blank correction is not considered, the concentrations of carbonyl compounds can be overestimated. In addition, if there is a dilution process in the sample pretreatment procedures, the error on the blank can be further increased. In this study, all concentrations of carbonyl compounds in HTP samples were determined with the correction of the system blank (system blank: cigarette smoking machine operated without installing an HTP device under HTP aerosol generation conditions; 40–68 ng/stick (FA), 608–786 ng/stick (AA), 55–242 ng/stick (ACR), 272–287 ng/stick (AT), 58–63 ng/stick (PA), 31 ng/stick (CA)). Hence, for these reasons, the concentrations of carbonyl compounds in HTP aerosols can differ significantly between studies (Table 3).

## 3. Materials and Methods

### 3.1. Generation, Sampling and Pretreatment of HTP Aerosol Samples

In this study, the possibility of producing harmful carbonyl compounds from heated HTP stick filters was examined. Three different HTP devices (different brands) were used, each with a representative HTP stick of the brand (HTP-1, HTP-2, and HTP-3). HTP aerosol was generated by heating the HTP sticks using the corresponding HTP devices. As the heating materials of the HTP sticks were different, three types of heating material samples were evaluated for each HTP brand-namely, the tobacco consumable without tobacco (sample A), part of the mouthpiece and the tobacco consumable without tobacco (sample B), and the tobacco consumable (sample C, conventional case) (Table 4 and Figure 3). All aerosols were generated under the “Health Canada Intense (ISO intense)” method [15]. Because the heating conditions of the HTP stick are not specified in the “Health Canada Intense” method (which is for conventional cigarettes), the HTP sticks were heated using HTP devices according to the manuals suggested by each manufacturer. In the case of samples A and C, although the heating duration time was very short (a total of 12 s), it was different from the heating method suggested in the manuals (manual = conventional method such as sample C) and there is a possibility that the heating temperature may be different from what would be seen using a conventional method. All samples were prepared and tested in triplicate.

The HTP aerosol samples (samples A, B, and C) of HTP-1, -2, and -3 were collected using a DNPH cartridge (Top Trading Co., Seoul, Korea) and a cigarette smoking machine (SG-300, Sibata, Soka-city, Saitama, Japan). The HTP aerosol generated by the HTP device heating system was transferred to the DNPH cartridge, which was equipped with an ozone scrubber (Top Trading Co., Seoul, Korea), using the cigarette smoking machine under the following puff conditions: (1) puff time = 2 s, (2) puff interval = 30 s, (3) puff volume = 55 mL, and (4) puff number = 6 (Table 4). The carbonyl-DNPH derivatives in the HTP aerosol samples collected in the DNPH cartridge were extracted using 5 mL of acetonitrile (Sigma-Aldrich, St. Louis, MO, USA). The extract was analyzed using an HPLC-UV system. To avoid the problems (unstable DNPH derivatization) associated with the unsaturated carbonyls such as acrolein and crotonaldehyde, the DNPH cartridge filled with a high concentration of DNPH was used and the ACN extraction was also performed with a sufficient time (above 5 min).

### 3.2. Preparation of Working Standard for Quantitative Analysis of Target Carbonyl Compounds

Six carbonyl compounds were selected as target analytes for this study, namely, formaldehyde (FA), acetaldehyde (AA), acrolein (ACR), acetone (AT), propionaldehyde (PA), and crotonaldehyde (CA) (Table 5). The primary standard consisted of six carbonyl-DNPH derivatives at concentrations of 15 μg·mL^−1^ (functional gravimetric concentration) (TO11A 6 Component Carbonyl-DNPH Mix, Supelco, St. Louis, MO, USA). The working standards for the five-point calibrations were prepared by dilution of the primary standard with acetonitrile to generate five different concentration levels (7.50, 15.0, 75.0, 150, and 300 ng·mL^−1^) for each carbonyl compound (Table 6).

### 3.3. Instrumentation

The carbonyl compounds in the working standards and sample solutions were analyzed using an HPLC system equipped with a UV detector (20A Series, Shimadzu, Kyoto, Japan). The working standards and sample solutions were placed in the HPLC autosampler and automatically injected into the HPLC column. The carbonyl compounds were separated on a Shim-Pack GIS-ODS column (length = 250 mm, diameter = 4.6 mm, and particle size = 5 μm) using a mobile phase of acetonitrile and distilled water at a flow rate of 1.5 mL·min^−1^ (total run time = 15 min). The separated compounds were detected using a UV detector at a wavelength of 360 nm. The detailed conditions are presented in Table 7.

## 4. Conclusions

It is well-known that several carbonyl compounds are generated by heating tobacco sticks for HTPs. When HTP sticks are heated using an HTP device, not only is the tobacco in the tobacco consumable part heated, but the filters in the mouthpiece part can be also be heated by heat transfer from the heated tobacco. In this study, the possibility of producing carbonyls from heated HTP stick filters was examined using various HTP sticks and their corresponding HTP devices. As different heating materials were examined, the generation of carbonyl compounds from the heated filters of HTP sticks could be confirmed.

When HTP aerosol was generated by heating the HTP stick using its device in the conventional way, FA was detected in the HTP aerosol samples (sample C) at a concentration of 0.641 ± 0.092 µg/stick. However, the HTP aerosol samples generated by heating some parts of the HTP mouthpiece without tobacco (sample B) contained higher FA concentrations (0. 945 ± 0.214 µg/stick). Although the method for the generation of the mouthpiece aerosols is not the conventional one and may have led to a different energy absorption during heating, these results show that more FA can be generated by heating the filter than by heating the tobacco. Similar to FA, ACR was also generated by the partial heating of HTP stick filters. In addition, AA and PA were detected in the HTP aerosol generated by the partial heating of HTP stick filters.

HTPs have been considered as alternatives to conventional cigarettes to reduce the health risks from harmful substances in cigarette smoke (or HTP aerosol). As previous evaluations have typically only analyzed and compared the harmful compounds in HTP aerosols and conventional cigarette smoke, HTP aerosol has been found to be preferential in terms of health risks. However, to achieve the effective regulation of HTPs, the generation of harmful compounds by heated tobacco stick filters in HTPs should be considered during safety evaluations of HTPs.

## Figures and Tables

**Figure 1 molecules-25-05612-f001:**
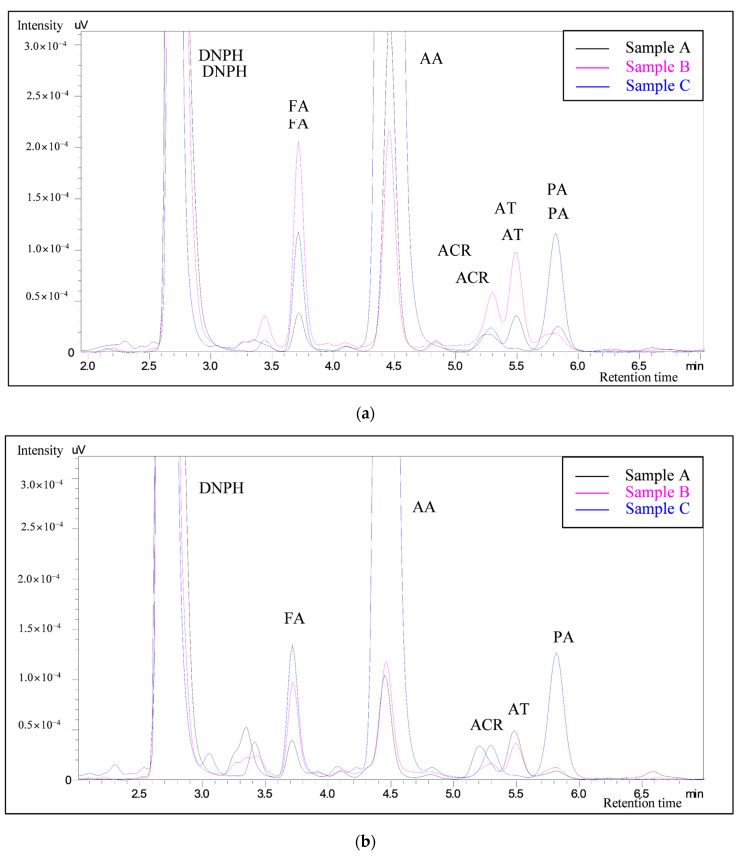
Chromatograms of the heated tobacco product (HTP) aerosol samples: (**a**) HTP-1; (**b**) HTP-2; (**c**) HTP-3.

**Figure 2 molecules-25-05612-f002:**
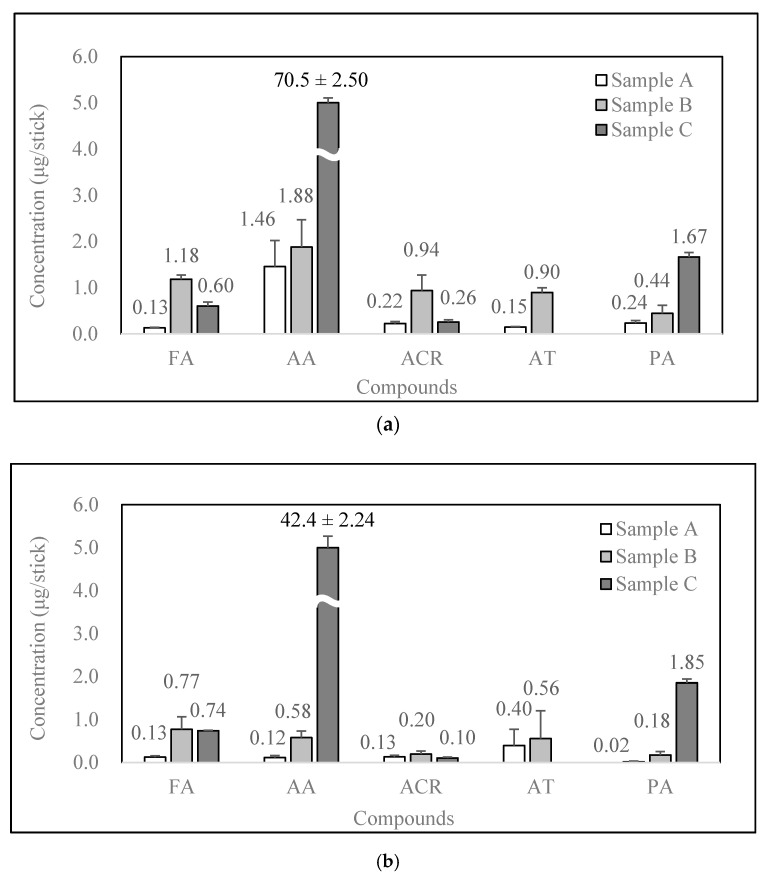
Comparison of the concentrations (μg/stick) of carbonyls in HTP aerosol samples: (**a**) HTP-1 (n = 3); (**b**) HTP-2 (n = 3); (**c**) HTP-3 (n = 3).

**Figure 3 molecules-25-05612-f003:**
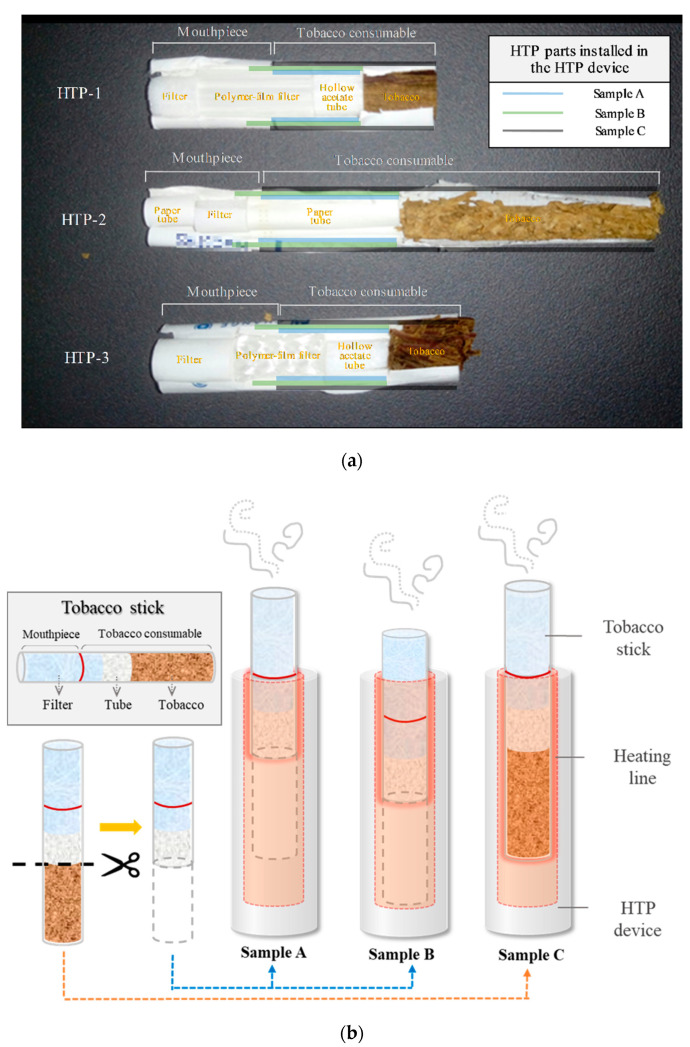
(**a**) Structures of tobacco sticks for HTP and (**b**) depictions of heating materials (samples A, B, and C).

**Table 1 molecules-25-05612-t001:** Calibration and quality assurance results for the working standards of six carbonyl compounds: (1) response factor (RF), (2) coefficient of determination (R^2^), (3) relative standard deviation (RSD), and (4) limit of detection (LOD).

Order	Parameters	Compounds ^d^					
		FA	AA	ACR	AT	PA	CA
1	RF (µL·ng^−^^1^)	521,019	405,507	357,263	294,963	304,518	243,030
2	R^2^	0.9999	0.9999	0.99997	0.9999	0.9999	0.9962
3	RSD ^a^ (%)	1.47	1.06	1.41	0.81	1.10	9.19
4	LOD ^b^ (solution: pg·µL^−^^1^)	0.106	0.136	0.155	0.187	0.182	0.227
5	LOD ^c^ (gas: ppbv)	1.31	1.15	1.02	1.20	1.16	1.20

^a^ Triplicate analyses of the 3rd calibration point (injection volume = 20 μL). ^b^ LOD was calculated using three times the standard deviation of background noise (n = 7). ^c^ Total sample volume: 0.33 L, temperature: 25 °C, analytical volume: 20 µL, and extraction volume: 5 mL for the derivation of molar ratio. ^d^ FA: formaldehyde, AA: acetaldehyde, ACR: acrolein, AT: acetone, PA: propionaldehyde, and CA: crotonaldehyde.

**Table 2 molecules-25-05612-t002:** ANOVA tests comparing the concentrations of target carbonyl compounds depending on heating materials and HTP brands.

	Grouping: Heating Materials			Grouping: HTP Brands		
Compounds	Sample Code	Concentration (µg/stick)	*p*-Value	HTP Brand	Concentration (µg/stick)	*p*-Value
	Sample A	0.138 ± 0.016		HTP-1	0.640 ± 0.528	
FA	Sample B	0.945 ± 0.214	9.05 × 10^−4^	HTP-2	0.546 ± 0.364	0.949
	Sample C	0.641 ± 0.092		HTP-3	0.539 ± 0.363	
	Sample A	0.616 ± 0.732		HTP-1	26.4 ± 42.9	
AA	Sample B	1.21 ± 0.650	5.05 × 10^−4^	HTP-2	14.4 ± 24.3	0.914
	Sample C	63.5 ± 18.4		HTP-3	24.5 ± 41.2	
	Sample A	0.121 ± 0.109		HTP-1	0.473 ± 0.402	
ACR	Sample B	0.519 ± 0.379	0.179	HTP-2	0.143 ± 0.047	0.358
	Sample C	0.220 ± 0.102		HTP-3	0.243 ± 0.214	
	Sample A	0.181 ± 0.200		HTP-1	0.348 ± 0.480	
AT	Sample B	0.580 ± 0.305	0.131	HTP-2	0.317 ± 0.286	0.540
	Sample C	Not available		HTP-3	0.096 ± 0.166	
	Sample A	0.102 ± 0.119		HTP-1	0.783 ± 0.771	
PA	Sample B	0.291 ± 0.139	8.54 × 10^−6^	HTP-2	0.682 ± 1.017	0.980
	Sample C	1.71 ± 0.123		HTP-3	0.641 ± 0.854	

**Table 3 molecules-25-05612-t003:** Comparison of the previous research data.

Previous Studies	Schaller et al. [11]	Forster et al. [12]	Farsalinos et al. [13]	Li et al. [14]	This Study (Sample C)
(a) Aerosol generation					
Device ^a^	THS2.2	THP1.0	IQOS	THS2.2	Three brands
Stick ^b^	FR1	T	NA ^c^	NA	3 types of sticks
Puff duration (sec)	2	2	2	2	2
Puff interval (sec)	30	30	30	30	30
Puff volume (mL)	55	55	55	55	55
Puff number (n)	12	8	12	12	6
(b) Concentrations of carbonyl compounds (μg/stick)					
FA	3.52 ± 0.3	3.29 ± 0.30	6.4 ± 1.8	21.9 ± 0.81	0.64 ± 0.092
AA	193 ± 2	111 ± 8	144 ± 23.3	210 ± 21.7	63.5 ± 18.4
ACR	9.76 ± 0.91	2.22 ± 0.52	10.8 ± 4.0	6.37 ± 0.32	0.220 ± 0.102
AT	37.7 ± 1.7	5.97 ± 0.66	NA	26.6 ± 1.17	ND ^d^
PA	14.4 ± 0.6	5.31 ± 0.15	12.8 ± 3.7	11.8 ± 0.38	1.71 ± 0.123
(n)	5	5	5	NA	3 × 3

^a^ THS2.2: Tobacco Heating System 2.2 (Philip Morris International), THP1.0: Glo (British American Tobacco). ^b^ FR1: THS2.2 Regular tobacco stick, T: Bright Tobacco Kent Neosticks. ^c^ NA: not available, ^d^ Not detected.

**Table 4 molecules-25-05612-t004:** Fundamental information about the samples and sampling conditions with different sampling approaches.

(a) Sample information						
	**Sample**					
**Sample Code ^a^**	**Target**		**Target Material ^a^**			
	**Product**					
Sample A	HTP-1, -2, and -3		Tobacco consumable without tobacco			
Sample B ^c^			Tobacco consumable without tobacco + Parts of mouthpiece			
Sample C			Tobacco consumable			
(b) Aerosol generation condition (based on “Health Canada Intense (ISO intense)” method)						
	**Aerosol Generation Condition**					
**Sample Code ^a^**	**Device**	**Puff**		**Puff**	**Puff**	**Puff**
	**Heating ^b^**	**Duration (sec)**		**Interval (sec)**	**Volume (mL)**	**Number**
Sample A						
Sample B ^c^	On	2		30	55	6
Sample C						

^a^ The target materials were directly contacted on the heating panel in HTP devices (Figure 3). ^b^ Each HTP device was used to heat the target materials of the samples. ^c^ The material was inserted farther into the HTP device than it would be in a conventional use case (Figure 3).

**Table 5 molecules-25-05612-t005:** List of six target carbonyl compounds in this study.

Order	Target	Short	MW	Density	Melting Point	Boiling Point	Formula	CAS
	Compounds	Name	(g·mol^−1^)	(g·cm^−3^)	(°C)	(°C)		Number
1	Formaldehyde	FA	30.0	0.815	−92	−19	CH_2_O	50-00-0
2	Acetaldehyde	AA	44.1	0.788	−123.5	20.2	C_2_H_4_O	75-07-0
3	Acrolein	ACR	56.1	0.839	−88	53	C_3_H_4_O	107-02-8
4	Acetone	AT	58.1	0.792	−95	56	C_3_H_6_O	67-64-1
5	Propionaldehyde	PA	58.080	0.81	−81	48	C_3_H_6_O	123-38-6
6	Crotonaldehyde	CA	70.091	0.846	−76.5	104	C_4_H_6_O	123-73-9

**Table 6 molecules-25-05612-t006:** Preparation of working standards containing target carbonyl compounds.

(a) The 1st working standards										
**Order**	**Mixing Volume (** **μ** **L)**		**Dilution**	**Concentration (ng·** **μ** **L^−1^)**					
	**PS ^a^**	**ACN ^b^**	**Fraction**	**FA**	**AA**	**ACR**	**AT**	**PA**	**CA**
1	200	1800	0.1	1.50	1.50	1.50	1.50	1.50	1.50
(b) The working standards at five concentration levels									
**Order**	**Mixing Volume (** **μ** **L)**		**Dilution**	**Concentration (ng·mL^−1^)**					
	**1st^-^WS**	**ACN ^b^**	**Fraction**	**FA**	**AA**	**ACR**	**AT**	**PA**	**CA**
1	10	1990	0.0050	7.50	7.50	7.50	7.50	7.50	7.50
2	20	1980	0.01	15.0	15.0	15.0	15.0	15.0	15.0
3	100	1900	0.050	75.0	75.0	75.0	75.0	75.0	75.0
4	200	1800	0.10	150	150	150	150	150	150
5	400	1600	0.2	300	300	300	300	300	300

^a^ Primary standard (PS) was purchased from Supelco (TO11A, USA): Concentration (functional gravimetric concentration) = each 15.0 μg·mL^–1^. ^b^ ACN: acetonitrile.

**Table 7 molecules-25-05612-t007:** Preparation of working standards containing target carbonyl compounds.

(a) Pump (LC-20AD)		
**Flow Rate**	1.5	mL·min^−^^1^
**Mobile Phase**	A: Distilled water	B: Acetonitrile
(b) Auto sampler (SIL-20A)		
**Injection Volume**	20	μL
**Mobile Phase**	0–4 min	A:B = 30:70
	4–8 min	A:B = 0:100
	8–15 min	A:B = 30:70
(c) Oven (CTO-20A)		
**Temp**	30	°C
**Column**	Shim-Pack GIS-ODS	
	(length: 250 mm, diameter: 4.6 mm, particle size: 5 μm)	
**Detected Time**	10	min
**Operation Time**	15	min
(d) UV detector (SPD-20A)		
**Wavelength**	360	nm

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
