# Peer review of "Carbonyl Compounds Containing Formaldehyde Produced from the Heated Mouthpiece of Tobacco Sticks for Heated Tobacco Products"

_molecules, 2020, doi:10.3390/molecules25235612_

Round 1

Reviewer 1 Report

The manuscript molecules-985704, entitled "Formaldehyde produced from the heated mouthpiece of tobacco sticks for heated tobacco products" presents the in-depth evaluation of the generation of carbonyl compounds from the various parts of currently available tobacco sticks, used as heated tobacco products. The authors detect several harmful products from the generated aerosols, among, which formaldehyde, acetone and acrolein were determined to be generated in higher concentrations due to the heat applied to the filter of the product.

The manuscript is well presented and would definitely be of interest for the readership of the journal. There are only a small number of issues and minor corrections that need to be addressed:

  1. The Materials and methods part should be put before the Results and discussion part. This would help the reader understand the various abbreviations and sample types, used in the study.
  2. The authors need to address the problems of determining acrolein using the DNPH cartridges. At least a few sentences should be dedicated to this issue. 
  3. Authors should change chromatogram B in Figure 1. See further information in the annotated pdf. 
  4. How many parallel measurements were taken? Please indicate the number of HTPs used for the determination of the average +/- SD.
  5. Other minor corrections are to be found in the annotated pdf. 

Reviewer 2 Report

In this paper the results from the determination of a number of carbonyl compounds in heated tobacco products is presented.  The novelty or originality of the presented work comes from the part of the tobacco product that is investigated.  This work focuses on the mouthpiece of the tobacco stick rather than the tobacco stick itself, and this is a worthwhile effort since (as the authors explain in the introduction) this part of the product gets heated as well.  Their findings thus suggest that the material that is used for the manufacturing of the mouthpiece should also be carefully inspected, which is valuable information for consideration in this field.  The paper is generally well written, but I do have some comments and suggestions for improvement and clarification that I will outline below:

  • The title should indicate that carbonyls (not just formaldehyde) are analyzed in this project, particularly since the top two compounds (in terms of concentration) were acetaldehyde and propionaldehyde.
  • In the abstract of the paper, the authors should delete the details of the 3 types of “samples” that they analyzed (sample A, B, and C), and instead briefly explain the relevance of analyzing these parts separately.
  • In the last paragraph of the introduction, I recommend the deletion of the words in parentheses (three prestigious brands).The sentence should just read as follows: “In this study, HTP aerosol was generated using three majorbrands of HTP sticks…”. The addition of the word “major” is a more appropriate description of the products selected for the study.
  • The “Materials and Methods” section of the paper should follow right after the introductory material is presented, before the “Results and Dicussion” section.
  • The authors should explain the aerosol producing protocol that is described in the Materials and Methods section. Is this protocol a standardized one?  The tobacco industry uses known standardized protocols (such as Coresta-81 for instance for e-liquid aerosol generation) in order to normalize the conditions under which results can be analyzed.  I understand that since the mouthpiece portion of the product is being analyzed, a strict adherence to Coresta-81 may not be possible. However, the authors should explain the reasons for their selection of aerosol generating parameters.  Otherwise the quantitative nature of their results cannot be effectively evaluated.  Alternatively, they could also provide information on the values of their aerosol producing protocol when used on the full product, in order to make a comparison to the amounts obtained on a known tobacco product when using the standardized method.
  • In Figure 1, five of the six carbonyls investigated in the study are identified. In the discussion of Figure 1 it is stated that crotonaldehyde (CA) was not found in any of the samples.  However, the wide peak that elutes early in the chromatograms is not identified.   A mention of this peak (is it DNPH “solvent”) should be done in the caption of Figure 1.

I think it is important for the authors to state (perhaps in the conclusion of the paper) that the mouthpiece aerosols generated in this study were not produced at the standardized conditions that are typically used. They can perhaps state that this would be an area for future work, since the results would then be able to be compared to amounts generated by other parts of the tobacco product.

Reviewer 3 Report

In this manuscript, the authors describe an attempt to determine the contributions of various tobacco stick mouth pieces to the carbonyl emission of heated tobacco products. This manuscript suffers from a number of methodological flaws: 

1. The carbonyl yields obtained during this study for sample C (normal use of the full stick) are not the same as those that have been previously reported in the literature by numerous authors (e.g. listed below), which clearly points to major methodological issues.

• Schaller et al. (2016) Evaluation of the Tobacco Heating System 2.2. Part 2: Chemical composition, genotoxicity, cytotoxicity, and physical properties of the aerosol. Regulatory Toxicology and Pharmacology, 81:S27-S34. • Forster et al. (2018) Assessment of novel tobacco heating product THP1.0. Part 3: Comprehensive chemical characterization of harmful and potentially harmful aerosol emissions. Regulatory Toxicology and Pharmacology 93:14-33. • Farsalinos et al. (2018) Carbonyl emissions from a novel heated tobacco product (IQOS): comparison with an e-cigarette and a tobacco cigarette. Addiction, 113:2099-2106. • Li et al. (2018) Chemical Analysis and Simulated Pyrolysis of Tobacco Heating System 2.2 Compared to Conventional Cigarettes. Nicotine & Tobacco Research, 21:111-118; 

It is generally advisable to validate measurement methods in comparison with published data.

2. The authors chose to remove the tobacco portion of the tobacco sticks and then heat the non-tobacco components of the sticks to determine the levels of 6 carbonyls emitted by these components. In doing so, the authors have completely changed the thermal properties of the sticks, which are likely to reach much higher temperatures than those in a complete and unaltered tobacco stick. Indeed, the heated tobacco products are designed in such a way that the heating element (be it a pin, a blade or a sleeve) provides the energy to the tobacco plug and thereby generates an aerosol. Cutting the tobacco plug will lead to a complete change in energy absorption patterns, and hence, the non-tobacco components of the sticks are exposed to thermal conditions that are fundamentally different from those occurring in a complete tobacco stick used under normal conditions.

In this context, the authors failed to provide temperature measurements at various locations within the sticks under all three experimental conditions. This makes it impossible to evaluate whether the experimental conditions are relevant to the normal use conditions. Importantly, cutting off the tobacco plug from the sticks most likely will cause overheating of the non-tobacco materials, which leads to their thermal degradation; something which does occur under normal product use conditions. 

Therefore, as it stands, the study described by the authors does not allow determining the contribution of the non-tobacco components to the carbonyl emissions under normal product use conditions. 

Furthermore, the authors fail to provide a scientific rationale, such as chemical reaction pathways, for the emission of carbonyls by the non-tobacco components of the sticks used under normal conditions.

Round 2

Reviewer 3 Report

The basic assumptions and approaches are flawed. I still do not quite understand the scientific rationale for conducting their research. The objective of this research is obscure to me.   Authors state:

28 [¨¨]Thus, safety
29 evaluations of heated tobacco products should include considerations of the harmful compounds
30 generated by heating tobacco stick mouthpieces for heated tobacco product in addition to those
31 found in heated tobacco product aerosols.

In fact this is where the whole paper falls down - the only important question for the evaluation of any heated tobacco product is what the consumer inhales. Hence only sample C is relevant.

The question on the thermal properties of samples A and B was not addressed, neither did they provide proper temperature measurements in those cases. This is obviously a critical point.
